# Quasi-continuous production of highly hyperpolarized carbon-13 contrast agents every 15 seconds within an MRI system

Andreas B. Schmidt [1,2,3✉], Mirko Zimmermann[1], Stephan Berner[1,2], Henri de Maissin [1,2], Christoph A. Müller [1,2], Vladislav Ivantaev[1], Jürgen Hennig[1], Dominik v. Elverfeldt [1] & Jan-Bernd Hövener [3✉]

Hyperpolarized contrast agents (HyCAs) have enabled unprecedented magnetic resonance imaging (MRI) of metabolism and pH in vivo. Producing HyCAs with currently available methods, however, is typically time and cost intensive. Here, we show virtually-continuous production of HyCAs using parahydrogen-induced polarization (PHIP), without stand-alone polarizer, but using a system integrated in an MRI instead. Polarization of $\approx$2% for [1-$^{13}$C] succinate-$d_2$ or $\approx$19% for hydroxyethyl-[1-$^{13}$C]propionate-$d_3$ was created every 15 s, for which fast, effective, and well-synchronized cycling of chemicals and reactions in conjunction with efficient spin-order transfer was key. We addressed these challenges using a dedicated, high-pressure, high-temperature reactor with integrated water-based heating and a setup operated via the MRI pulse program. As PHIP of several biologically relevant HyCAs has recently been described, this Rapid-PHIP technique promises fast preclinical studies, repeated administration or continuous infusion within a single lifetime of the agent, as well as a prolonged window for observation with signal averaging and dynamic monitoring of metabolic alterations.

[1] Department of Radiology, Medical Physics, Medical Center, Faculty of Freiburg, University of Freiburg, Killianstr. 5a, Freiburg 79106, Germany. [2] German Cancer Consortium (DKTK), partner site Freiburg and German Cancer Research Center (DKFZ), Im Neuenheimer Feld 280, Heidelberg 69120, Germany. [3] Section Biomedical Imaging, Molecular Imaging North Competence Center (MOIN CC), Department of Radiology and Neuroradiology, University Medical Center Schleswig-Holstein, Kiel University, Am Botanischen Garten 14, 24118 Kiel, Germany. ✉email: andreas.schmidt@uniklinik-freiburg.de; jan.hoevener@rad.uni-kiel.de

Magnetic resonance (MR) has revolutionized many fields, including analytical chemistry and medical diagnostics. These successes were made despite the fact that MR uses only a tiny fraction of the theoretically available signal: No more than a few parts per million of all spins in a sample effectively contribute to the signal in magnetic fields of a few tesla and at room temperature. Hence, clinical MR imaging (MRI) is limited to accessing the most abundant substances in the body, i.e., water and lipids[1], with few exceptions[2,3]. MRI of hyperpolarized contrast agents (HyCAs) has enabled hitherto unprecedented imaging of airspaces, lung function and metabolism[4–9]. The widespread application of this technique, however, was hindered thus far by complex, expensive hardware, and long preparation times: Using the current standard technique, i.e., dissolution dynamic nuclear polarization (dDNP), typically more than an hour is needed to produce a single dose of $^{13}$C hyperpolarized metabolites with a stand-alone device (polarizer) costing more than a million euros. Still, the need to produce multiple samples fast was recognized and resulted in the development of great technologies[10–12]. These advances included the spinlab polarizer, which allowed polarizing several samples at the same time, but required an overnight recovery period; the spin-aligner dDNP polarizer ingeniously overcame this drawback allowing a remarkable duty cycle of about one polarized sample, e.g., every hour[13,14].

Parahydrogen (pH$_2$) induced polarization (PHIP) methods are drastically less expensive and much faster than most other hyperpolarization (HP) schemes[15–19]. There are two variants of PHIP: one where pH$_2$ is catalytically added to a precursor (referred to as hydrogenative PHIP, parahydrogen and synthesis allow dramatically enhanced nuclear alignment (PASADENA)[15,20] or adiabatic longitudinal transport after dissociation engenders net alignment (ALTADENA)[21]), and another where the substrate undergoes a reversible exchange with pH$_2$ at a catalyst and thus remains chemically unchanged (non-hydrogenative PHIP, signal amplification by reversible exchange (SABRE))[17,22,23]. For the former, duty cycles of a few minutes were achieved[24–26] – where the actual polarization, consisting of hydrogenation and spin order transfer (SOT), took <10 s at elevated temperatures and pH$_2$ pressures – e.g., by preparing several (frozen) samples in nuclear magnetic resonance (NMR) tubes[27–32], or amounts of precursor-catalyst solution to be used in PHIP polarizers for production of agents polarized to >10%[24,25,33–39]. Such batches of PHIP-polarized carbon-13 HPCAs have recently been used for metabolic imaging with pyruvate[8] and fumarate[40].

A very interesting development is continuous HP[41], e.g., using excess of precursor and continuous pH$_2$ supply[42] or continuous flow setups, e.g., to be used in lab-on-a-chip NMR devices[43–46]. Only recently, continuous-flow PHIP of an acetate ester was demonstrated by bubble-free pH$_2$ dissolution and heterogeneous catalysis, which makes production of catalyst-free solutions straightforward[47]. SABRE is particularly well suited for continuous HP, because it does not consume the precursor[41,48–52]. Progress is being made fast and continuous polarization of $^1$H and X-nuclei (nuclei other than proton, e.g., $^{15}$N, $^{13}$C) in metabolites to ≈6% and ≈2%, respectively, was recently demonstrated in a bubble-free membrane reactor[53].

For biomedical use, typically, high concentrations, high polarizations >10%, a long polarization lifetime of ≈min, and little or no background signal in vivo are needed. These needs were met by implementing hydrogenative PHIP within the bore of an MRI system – a method referred to as synthesis amid the magnet bore allows dramatically enhanced nuclear alignment (SAMBADENA)[36,54,55]. The method allowed to acquire hyperpolarized MRI in vivo within seconds after polarization at less

than 1% of the cost of the currently available dDNP polarizers (using the hardware of the MRI system, and disregarding the cost of the pH$_2$ generator)[56].

Here, we present the quasi-continuous production of $^{13}$C HyCAs polarized by SAMBADENA within the MRI magnet. With a new setup, a fast hyperpolarization, and cycling of the HyCAs and precursors, we polarized agents repeatedly every 15 s with mean polarizations of up to 20%. This duty cycle is several times faster than the longitudinal relaxation time (for $^{13}$C often ≈1 min) and holds great potential for new applications of hyperpolarized MRI with continuous infusion or repeated injections within the lifetime of an agent.

## Results

**Rapid parahydrogen induced polarization.** In contrast to previous work, where approximately two minutes were required to repeatedly produce SAMBADENA-hyperpolarized HyCAs[56], we were able to produce highly $^{13}$C-polarized samples every 15 s (Fig. 1). Using the vast spin-order reservoir of parahydrogen in conjunction with a fast chemical reaction[24,57,58] and rapid spin order transfer scheme[59–61], each batch was produced much faster than the typical lifetimes of $^{13}$C HP, e.g., ≈75 s for the agent hydroxyethyl [1-$^{13}$C]propionate-d$_3$ (HEP; Supplementary Note 1 and Supplementary Fig. 4). We chose to demonstrate this technique using two well-known, commercially available PHIP agents that have been established for SAMBADENA: HEP[25,33,37,62,63] and [1-$^{13}$C]succinate-d$_2$ (SUC)[37,64–67]. Every 15 s, a 700 μL batch containing 5 mM HEP or SUC was polarized for 10 and 9 times in a row, respectively. The mean $^{13}$C polarization was quantified with respect to a thermally polarized reference (carbonyl resonance of 5 mL neat acetone, natural abundance $^{13}$C of ≈1.1%) to $P_{HP}(HEP) = (19 ± 1)\%$ and $P_{HP}(SUC) = (1.7 ± 0.2)\%$ (mean ± standard error; the standard deviation in absolute numbers was

[1-$^{13}$C]HEP-d$_3$
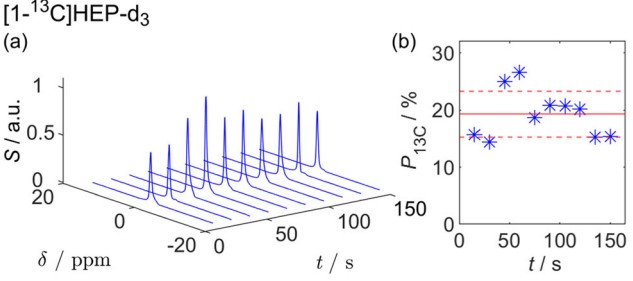

[1-$^{13}$C]SUC-d$_2$
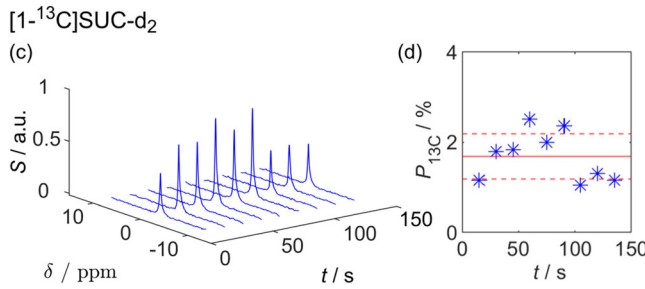

**Fig. 1 Producing batches of $^{13}$C hyperpolarized contrast agents every 15 s.** NMR spectra (**a**, **c**) and corresponding $^{13}$C polarization (**b**, **d**; asterisks) of hyperpolarized HEP and SUC. The mean $^{13}$C polarization (red line in **b** and **d**) with standard error, $s_E$, was $P_{13C} = (19 ± 1)\%$ and $P_{13C} = (1.7 ± 0.2)\%$, respectively (standard deviation of all N experiments, $\sigma = \sqrt{N} \cdot s_E$, plotted as red dashed line). The concentration of catalyst and precursor were 2 mM and 5 mM in deionized H$_2$O, respectively. For each sample, hydrogenation was carried out for 5 s at 15 bar and 80 °C. Data is reported in Supplementary Table 2.

±4% and ±0.5%, respectively; Fig. 1). The measured mean polarizations correspond to signal enhancements of ≈31,700- and ≈2800-fold with respect to the equilibrium polarization at 7 T. These numbers require some commenting: First, the polarizations are likely underestimated as we assumed complete hydrogenation, whereas about ≈90% were actually achieved (see below); complete hydrogenation would increase the polarization by a factor of ~1.11. Second, using $pH_2$ enriched to 100% (instead of 85% used here, Supplementary Fig. 1), would increase the polarization by another factor of ≈1.25. Third, we note that the standard deviation of rapid-PHIP (±20%) is larger than for single-batch SAM-BADENA (±10%). We attribute this variation in $P_{13C}$ to varying volumes of HPCA in the reactor, whereas we assume a constant volume (700 µl) for quantification. We observed some variation in volume caused by the fast injection of unreacted precursor solution into the reactor, and to some spill over during the hydrogenation reaction.

**Hyperpolarization procedure.** The actual PHIP process, including hydrogenation of the precursor with $pH_2$ (Fig. 2a) and SOT, took only 5 s per sample (HEP: using the $pH_2$ insensitive nuclei enhanced by polarization transfer (phINEPT+)[59]; SUC: using Goldman's sequence)[68]. The remainder of the 15-s duty cycle was used to empty (5 s) and refill (5 s) the reaction chamber with the aqueous catalyst-substrate solution (Fig. 2b). The precise experimental procedure, chemicals, setup, and quantification are reported in the methods and SI. Note that polarization for HEP maybe doubled (i.e., to ≈40% for $^{13}C$) if a more efficient SOT method would be used (that provides up to 100% polarization

instead of 50% used here)—a matter currently being investigated[61,69,70]. Also for SUC, other SOT sequences (i.e., for strongly-coupled spin systems) seem to be well-suited like the (generalized) S2hM sequence and should be tested in future experiments[54,55,60,71–76].

For rapid iteration of the experiment, it was key to achieve a precise synchronization of the MRI and the hyperpolarization hardware as well as a fast formation of the contrast agent (within 5 s throughout all repetitions of the hyperpolarization cycle). To meet these challenges, we developed a new setup (Fig. 2c), where the entire process was controlled by the MRI system on a 12.5 ns time grid. The digital outputs of the MRI were used to send commands to a preprogrammed microcontroller which, in turn, was used to switch the electromagnetic valves via power relays to handle the fluids (Supplementary Fig. 3). To assure a fast hydrogenation, we developed a reaction chamber (PSU 2000) that allowed for high reaction pressures and integrated water-based heating (Supplementary Fig. 2). The stock of aqueous catalyst-precursor solution was kept at the back of the MRI system in a heated water bath (80 °C) and was transferred into the reaction chamber via a thin tube as needed. This way, a high pressure of $pH_2$ (≥15 bar) and high reaction temperature (≈80 °C) were realized and maintained throughout all repetitions. At the given timepoint, the $pH_2$-derived spin order was transferred to $^{13}C$ by playing out the SOT pulse sequence using a dual-tune, linearly polarized $^1H$-$^{13}C$ coil and hyperpolarized signal was detected at the end of the sequence.

**Fast hydrogenation reaction.** This new setup was used to optimize rapid PHIP (Fig. 3). Not surprisingly, we found that the reaction temperature was beneficial for the HP yield, just as well as the $pH_2$ pressure (Fig. 3a, b[25,36]—note that the hydrogenation yield was not considered in the quantification, but assumed to be 100%). The catalyst concentration did not have a pronounced effect on the polarization in the range of 1–4 mM for 80 mM of HEA before (Fig. 3c)[56]. As expected, the $^{13}C$ polarization decreased and the payload (precursor concentration multiplied by polarization) reached a maximum when the concentration of the precursor hydroxyethyl [1-$^{13}C$]acrylate-$d_3$ (HEA), $c_{HEA}$, was increased from 5 mM to 80 mM under otherwise same conditions (Fig. 3d). A mono-exponential saturation function fitted the payload (d) as function of $c_{HEA}$ well, suggesting that one other reactant ($pH_2$, or catalyst) was used. Together with (b) and (c), this observation suggests that $pH_2$ was likely the rate limiting reactant here, and that the saturation of the payload in Fig. 3d was attributable to the consumption of the available $pH_2$. Prolonging the hydrogenation period increased the hydrogenation yield, but caused relaxation at the same time[36]. Thus, there was an optimum hydrogenation time, $t_h$, of 5 s for which the polarization, under the chosen parameters, was maximal (Fig. 3e).

In some cases, however, reducing the amount of unreacted precursor may be desired, in particular if it is toxic. While this could be addressed with a purification of the solution, e.g., with precipitation[39,77], a longer or faster hydrogenation may solve this issue, too. To investigate the matter further, a two-handed kinetics model has been fitted to the polarization as function of $t_h$ (Fig. 3e):[32,36,55,78]

$$P_{13C}(t_h) = P_{max} / \left| T_{hydr}/T_{relax} - 1 \right|$$
$$\cdot \left[ \exp\left(-(t_h - t_0)/T_{hydr}\right) - \exp\left(-(t_h - t_0)/T_{relax}\right) \right]$$

where $T_{hydr}$ and $T_{relax}$ are time constants of the hydrogenation and relaxation of the $^1H$ spin order, respectively, $P_{max}$ is the maximum polarization value (reached if no relaxation were present), and $t_0$ is a time offset (delivery and dissolution

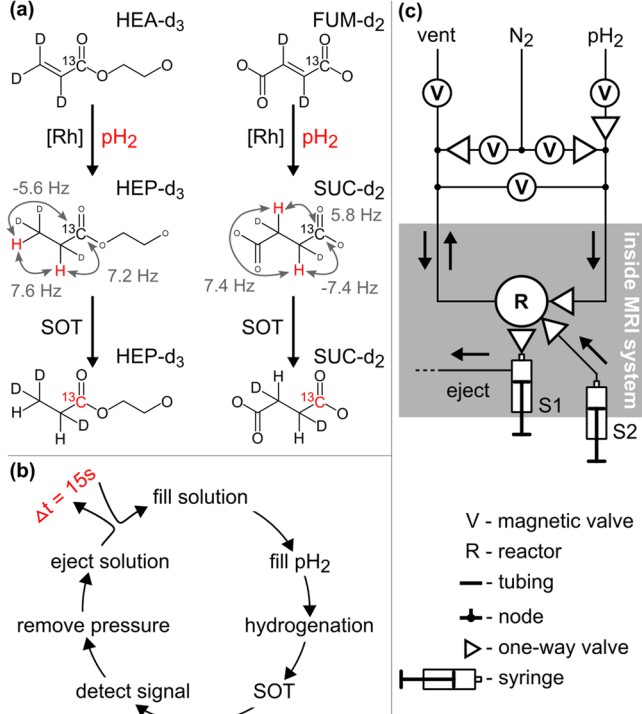

**Fig. 2 Schematics of rapid PHIP.** Polarization procedure including $pH_2$ addition and SOT (**a**), the experimental cycle (**b**), and a diagram of the experimental setup (**c**). In (**a**), hydrogenation of a precursor molecule forms the $^1H$-hyperpolarized product (J-couplings between $^1H$ and $^{13}C$ nuclei indicated)[65,68]; subsequently, spin order is transferred to 1-$^{13}C$ via J-couplings and a pulse sequence (polarized nuclei are indicated in red). A more detailed presentation of the experimental procedure and setup (**c**) with technical drawings and photographs is given in the SI.

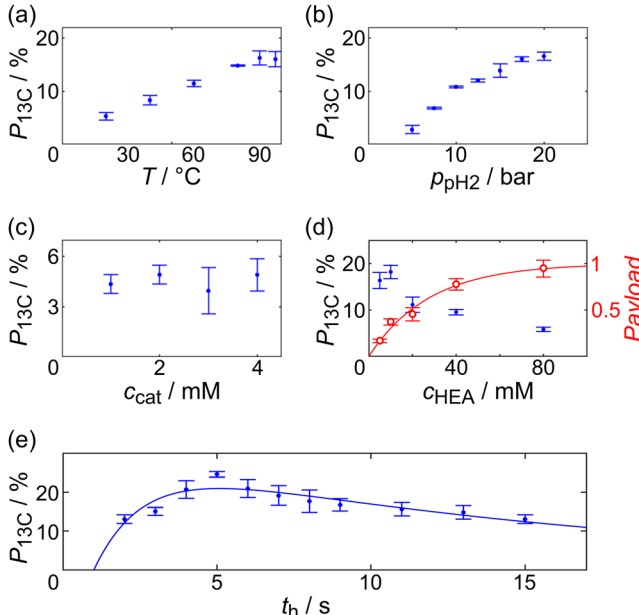

**Fig. 3 Optimization of rapid PHIP.** $^{13}$C hyperpolarization $P_{13C}$ of HEP (blue dots) as a function of temperature $T$ (**a**), parahydrogen pressure $p_{pH2}$ (**b**, $T = 90\,°C$), concentration of catalyst $c_{cat}$ (**c**, $c_{HEA} = 80$ mM), precursor $c_{HEA}$ (**d**), and hydrogenation time $t_h$ (**e**, $c_{HEA} = 5$ mM). In (**d**), the payload ($c_{HEA} \cdot P_{13C}$; normalized; red circles) is also shown with errors calculated from errors of $P_{13C}(c_{HEA})$ using Gaussian error propagation. Fitted curves correspond to a mono-exponential saturation function (**d**; $R^2 = 0.98$) and the hydrogenation kinetics model from the main text (**e**; $R^2 = 0.7$). Unless stated otherwise, parameters were $T = 80\,°C$, $p_{pH2} = 15$ bar, $c_{cat} = 2$ mM, $c_{HEA} = 20$ mM, $t_h = 5$ s, and the mean and standard error of $N = 3$ experiments is reported; $P_{13C}$ was quantified assuming hydrogenation of the full amount of precursor was completed. Data is reported in Supplementary Table 3. Data in (**c**) and (**e**) were published before but add to the matter at hand[36,56].

of $p$H$_2$, build-up of pressure, and activation of catalyst). The fitted parameters were: $T_{hydr} = (1.6 \pm 0.9)$ s, $T_{relax} = (16 \pm 6)$ s, $P_{max} = (27 \pm 5)\%$, and $t_0 = (1.0 \pm 0.5)$ s. With $f(t_h; t_0, T_{hydr}) = \exp(-(t_h - t_0)/T_{hydr})$, the reacted fraction, $f$, with propagated error, $s_f$, after hydrogenation time $t$ can be calculated as

$$f \pm s_f = f\left(t_h; t_0, T_{hydr}\right) \pm \left(\left(\frac{\partial}{\partial t_0} \cdot f\left(t; t_0, T_{hydr}\right)\right)^2 \cdot s_{t_0}^2 \right.$$
$$\left. + \left(\frac{\partial}{\partial T_{hydr}} \cdot f\left(t; t_0, T_{hydr}\right)\right)^2 \cdot s_{T_{hydr}}^2\right)^{1/2}.$$

Hence, the model suggested that after 5 s, $(92 \pm 12)\%$, and after 10 s, $(99 \pm 1)\%$ of the precursor were reacted (under the given conditions of $p_{pH2} = 15$ bar, $c_{cat} = 2$ mM, $c_{HEA} = 5$ mM).

Combined, these findings suggest that a faster reaction and an increased $p$H$_2$ availability would improve the hydrogenation and allow to effectively polarize higher concentrations (e.g., 80 mM, Fig. 3d). Increasing the pressure is one solution that we are currently pursuing. Another one is to use solvents with higher $p$H$_2$ solubility than water ($\approx 0.8$ mM bar$^{-1}$ at $25\,°C$)[79,80], for instance chloroform ($\approx 2.7$ mM bar$^{-1}$ at $25\,°C$)[81] or acetone ($\approx 3.9$ mM bar$^{-1}$ at $25\,°C$)[82]. Note that these solvents are typically used for PHIP by side arm hydrogenation (PHIP-SAH)[83] for production of metabolic PHIP agents, which makes this approach most promising, too. Interestingly, the solubility of H$_2$ in chloroform and acetone increases with temperature (e.g., to

3.4 and 4.6 mM bar$^{-1}$ at $50\,°C$, respectively), whereas it decreases for water (0.7 mM bar$^{-1}$ at $50\,°C$)[79,81,82,84].

**Adaption of the temperature of the HyCA.** Encouraged by these results, we tested if the contrast agent could be rapidly cooled to a level suitable for in vivo applications. We found that the temperature of ten 500 μL doses subsequently ejected at a flow rate of $\approx 100$ μL s$^{-1}$ was reliably reduced to $(32 \pm 2)\,°C$ at the tip of the catheter when $\approx 5$ cm of the tube was guided through a reservoir of $16\,°C$ water (Supplementary Note 2 and Supplementary Table 1; inner diameter of the catheter was 0.28 mm). Thus, rapid adjustment of the temperature is feasible, and the agent is available for infusion or repeated injections every 15 s.

## Discussion

Producing four boli of hyperpolarized contrast agents per minute (or 10 in 2.5 min) as demonstrated here is about two orders of magnitude faster than reported for dDNP, where one sample was produced every hour[13], or where up to four samples were simultaneously polarized in ~ 3 h, and dissolved in intervals of 12 min, followed by a 30-h recovery period[12]. Compared to duty cycles of minutes reported for hydrogenative PHIP to produce HP batches of $P_{13C} > 10\%$, a 15-s duty cycle is still several times faster, e.g., by a factor of $\approx 12$[37] or $\approx 20$[24]. Moreover, producing hyperpolarized agents faster than the relaxation is particularly interesting as, in some aspect, the fierce limitations imposed by $T_1$ are ameliorated: It appears feasible to inject, or actually infuse hyperpolarized agents over a prolonged period of time and to perform signal averaging. Given that the clinical use of hyperpolarized MRI is very strongly linked to the (usually limited) $^{13}$C signal, this approach may prove to provide just the extra signal-to-noise ratio needed (disregarding on how the contrast agents are hyperpolarized).

SAMBADENA, specifically, allows the production and application of agents in the MRI system, reducing the relaxation between hyperpolarization and administration[36,56]. Of course, purity and quality would need to be assured prior to any injection in vivo. However, great progress was made recently and a purification within tens of seconds during the hyperpolarization of the next batch seems feasible[39,58,85,86].

In this context, it is reasonable to consider the limits of PHIP and hyperpolarization in general: For all practical matters, $p$H$_2$—as a reservoir of spin order—has a virtually infinite capacity to polarize contrast agents. A human dose will not require more than ~10 mmol of contrast agent in the injection solution (e.g., 40 mL with 250 mM)[87]. $p$H$_2$ can be stored for days and produced at rates of ~1 mol s$^{-1}$[88–93]. The steps limiting the production of hyperpolarized agents are rather the solubility of H$_2$ in solution[79,80,82], as well as the chemical reaction, which incorporates the spin order into the target agent (Fig. 3e)[36]. Both issues, however, can be tuned to a degree that allows virtually continuous production of hyperpolarized agents much faster than their typical lifetime, as was demonstrated in this work. What remains to be solved are suitable agents and a production that is safe for animal or human applications, tasks that appear feasible but shall not be underestimated.

To exemplify this technique, we chose HEP and SUC, two PHIP agents that were used for >10 years. Extension of the method to more recent developments such as PHIP by side-arm hydrogenation (PHIP-SAH)[83], and producing metabolically active agents like fumarate[39,40,94,95], or (esters of) pyruvate[8,27,28,83,96,97], acetate[29,30,32,83,98], lactate[99–101], appears feasible but will require some technical changes in the setup (e.g., solvent compatibility). Likewise, filtering or capturing the catalyst[58,86,102] and using heterogeneous catalysis[47,103–106]

appears to be amenable for this technique, too. Obviously, realization of these points will require more research.

Producing highly hyperpolarized agents within a few seconds, reproducibly, and repeatedly, opens up new prospects for MRI of hyperpolarized contrast agents. These may be used for serial injections, hyperpolarized infusion, signal averaging and prolonged scanning; - an area of research that we open with this contribution.

## Methods

The experimental setup consisted of the MRI system used for signal detection and SOT and the HP setup, comprising the fluid control and the reactor.

**MRI.** A 7T preclinical MRI system (Biospec 70/20, PV5.1, Bruker, Germany) was used along with a dual-resonant $^1$H-$^{13}$C volume transmit-receive coil with a length of 10 cm and an inner diameter of 7.2 cm (V-XLS-HL-070-01349, Rapid, Germany) for HP and signal acquisition.

**Hyperpolarization setup – reactor.** A polysulfone reactor was custom designed and built to allow for a stable, elevated temperature and high pressure throughout the rapidly repeated hyperpolarization experiments, thus ultimately, to allow for a fast hydrogenation reaction (Supplementary Fig. 2; inner volume of ≈4 ml, PSU 1000). The inner reaction chamber had four connections for filling and removing reaction solution and pressurized gases ($N_2$ and $pH_2$). The reaction chamber was surrounded by a second chamber at a distance of 2 mm, through which hot water (90–95 °C) was guided during the experiments using silicone tubing (6 mm OD × 4 mm ID, BONI Schlauch und Schlauchtechnik, Germany) and a water pump (1P T12184, Thermo Fisher Scientific, Newington, NH, USA). Note that the tubing was insulated using radiator insulation outside the MRI bore (43 mm OD × 13 mm ID, Insul-Tube HiTemp, NMC, Belgium) and a sleeving inside the MRI system (ID of 0.263", FGLG.02BK, Techflex, Germany). While the pressure in the experiments was not higher than 20 bar, it may be interesting to note that the reactor was designed for applications with much higher pressure: Finite element simulation (Inventor Professional 2018, Autodesk, USA) suggested that the reactor resists up to 273 bar in the inner chamber without permanent deformation. Thus, a pressure up to 91 bar can be loaded with a safety factor of 3, with the weakest element being the reactor lid. Experimentally, 50 bar were loaded statically or dynamically without damaging the reactor. Similar to a previous setup[56], the reactor was combined with a custom-made animal bed, which was mounted to a motorized slider (AutoPac, Bruker, Germany), allowing exact positioning of the reactor in the isocenter of the magnet.

**Hyperpolarization setup – fluid control.** Flow of gases ($N_2$ and $pH_2$) and the solution was controlled from the pulse program of the MRI system via its transistor-transistor logic (TTL) outputs and a dedicated, microcontroller-based setup (Teensy 3.5, Arduino; Supplementary Fig. 3). The latter comprised five electronically switchable magnetic valves (G052.00, GSR Ventiltechnik, Germany), and five one-way in-line valves that directed flow (two outside the magnet, SS-4CP2-RT-25, Swagelok, USA; three inside the magnet mounted to three of the connections of the reactor, Non-Metallic Check Valves CV33 series, IDEX Health & Science LLC, USA). The system enabled filling and emptying the reactor with the solution while it was positioned inside the MRI system, which allowed the virtually-continuous hyperpolarization. To avoid contamination of the electromagnetic valves, a custom-made fluid trap was included into the outlet path of the reactor. The catalyst-substrate solution was filled from a 10 ml syringe (Luer-Lok Tip, BD, USA) through an ETFE tube (1/16" OD x 0.5 mm ID, 1 m length, ~0.2 mL inner volume, SCP, Germany). The reacted solution was removed from the reactor via a modified syringe, which was combined with another tube (OD of 4 mm, ID of 2 mm, non-magnetic, TPE-U(PU), Festo, Germany) to guide the polarized solution into a container also placed posterior to the magnet.

**Experimental procedure.** The reactor was warmed for ≈5 min to ≈80 °C by operating the integrated heating chamber. Then it was filled with 1 mL, ≈80 °C $H_2O$, and positioned in the isocenter of the MRI system to perform the adjustments of the static magnetic and $^1$H-radiofrequency field (1-$^{13}$C reference power was known from previous experiments). Right before the experiment, a batch of 10 mL catalyst-precursor solution was warmed in a 10 mL polycarbonate syringe (Luer-Lok Tip, BD, USA, ≈80 °C) that was connected to the solution inlet of the reactor. The experiment was triggered by the operator by pushing a button on the backside of the MRI system. Then, the HP procedure—that was fully automated except for the two syringes (S1 and S2 in Supplementary Fig. 3), which were operated manually—was executed and repeated for several times every 15 s. The stepwise procedure and operation of the valves is detailed in the Supplementary Methods.

Additional details on the materials and methods, including preparation of parahydrogen and samples, the quantification of HP and pulse programs (Paravision 6.1, Bruker, Germany) are presented in the Supplementary Methods.

## Data availability

The authors declare that the data supporting the findings of this study are available within the paper and Supplementary Tables 1–4. Raw data are available from the corresponding author upon request.

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

## Acknowledgements

We acknowledge funding from the German Cancer Consortium (DKTK), DFG (SCHM 3694/1-1, HO-4602/2-2, HO-4602/3, GRK2154-2019, EXC2167, FOR5042, SFB1479, TRR287), Kiel University and the Faculty of Medicine, and Research Commission of the University Medical Center Freiburg (SCHM2146-20). MOIN CC was founded by a grant from the European Regional Development Fund (ERDF) and the Zukunftsprogramm Wirtschaft of Schleswig-Holstein (Project no. 122-09-053). We thank the Core Facility Advanced Molecular Imaging Research Center (AMIR), Department of Radiology – Medical Physics of the University Hospital Freiburg for support in acquisition and analysis of the data.

## Author contributions

A.B.S., J.B.H.: conceptualization, methodology, writing original draft, preparation, funding acquisition. A.B.S., M.Z., S.B., H.d.M., C.M., V.I.: investigation. J.H., D.v.E., and J.B.H.: supervision. All authors contributed to discussions, helped interpreting the results, and approved the final version of the manuscript.

## Funding

## Competing interests

The authors declare no competing interests.
