## [Peer Review File · Communications Chemistry]

Reviewers' comments:

Reviewer #1 (Remarks to the Author):

The contribution by Schmidt et al. is an exciting step forward for the field of MRI using PHIP-polarized metabolites. The authors report on a new method to generate hyperpolarized biomolecules with a turnover time of 15s, which is close to continuous production. The achieved polarization levels are in line with the current state-of-the-art for the two molecules HEP and SUC, but the turnover rate is much higher. This opens the door to quasi-continuous or continuous hyperpolarized imaging.

Overall I think this work is well-suited for publication in Communications Chemistry, subject to a few minor corrections (see below).

Some specific comments to the authors:

1. In the introduction I think some more care should be given to introducing the current state-of-the-art in generating PHIP-polarized molecules in a quasi-continuous or continuous manner. The reader is not made aware of the current limitations in PHIP technology, and what advance has been made in this work. No mention is given to work in which many NMR tubes can be prepared containing frozen PHIP precursor solutions and used in quick succession (e.g., <https://doi.org/10.1016/j.jmr.2018.01.019>). Also, there is no explicit discussion of bubble-free PHIP setups which achieve continuous production of PHIP-polarized molecules (albeit at lower volume/concentration, which is worth discussing). See for example: <https://doi.org/10.1002/anie.201002725>, <https://doi.org/10.1021/jacs.9b03507>, <https://doi.org/10.1002/cphc.202100119> (which is already cited but lumped together with very different work).
2. Line 42: Duty cycle is the wrong parameter in this context. Turnover time (or similar) would be better.
3. Line 64: "The deviation of the signals was likely dominated by the varying amount of filled reaction solution." I do not understand this: why should this have such a large impact on polarization, and why is there such a significant standard deviation on the amount of reaction solution being used? I think this point certainly needs clarifying, since the standard deviation of polarization looks close to 20% in both cases.
4. Line 78: I think there are suitable pulse sequences for this such as S2hM (<https://doi.org/10.1016/j.jmr.2017.03.002>) and even generalized-S2hM (<https://doi.org/10.1016/j.jmr.2020.106850>). I don't insist on repeating experiments using these pulse sequences, but they could be mentioned as viable alternatives.
5. Line 127: I take issue with the statement "Producing four boli of hyperpolarized contrast agents per minute is at least two orders of magnitude faster than reported before.". Many PHIP laboratories are able to repeat experiments faster than once every 25 minutes. This is even stated on line 42 in the Introduction. I think one order of magnitude is more accurate.

Reviewer #2 (Remarks to the Author):

This is a review for a manuscript recently submitted by Schmidt et al. entitled "Quasi-continuous production of highly hyperpolarized carbon-13 contrast agents every 15 s within an MRI system",

submitted for consideration for publication in Communications Chemistry.

Ultimately, I believe the paper will be publishable once some relatively minor issues and questions have been addressed (mostly just concerning how the authors frame some of their advances), given that most of the science itself seems to be in good order (and the paper for the most part is clearly written).

In their Communication, the authors describe recent work to develop a rapid and reproducible approach for creating select hyperpolarized ^{13}C -labeled agents for biomedical imaging applications via parahydrogen-induced polarization (PHIP). The approach exploits methodologies previously developed by the group, largely under the acronym SAMBADENA--a PHIP process that occurs within an MRI scanner (facilitating the rapid transfer of hyperpolarized agents to the target/subject with minimal relaxation losses). In addition to rapid agent production, good reproducibly and relatively high ^{13}C polarizations are obtained by optimizing this process (e.g. $P(^{13}\text{C}) \sim 2\%$ for succinate or $\sim 19\%$ for hydroxyethyl-propionate, with batches every 15 s) with rational experimental design centered around a dedicated PHIP reactor, which should ultimately help enable PHIP-enhanced metabolic MRI with continuous (or at least quasi-continuous) HP agent administration.

Specific comments / questions / corrections (in rough order of the manuscript):

- In many places, the authors say that they achieve their results “without a polarizer”. This strikes me as a strange claim: (1) the authors make it sound like this is somehow an advantage, but it’s not clear to me why this would be an advantage as one must have *some* way to make HP agents; and (2) in fact, I would say that the authors do indeed have a “polarizer”. The authors describe a dedicated heated/controlled PHIP reactor with additional controls on the front and back end (including integration with the MRI scanner), such that into it goes pH_2 and substrate, and out comes HP agent (quite nicely). Perhaps it does not have a fully enclosed cabinet and wheels, but besides that, it sounds like a PHIP polarizer to me! I suggest the authors either delete this line of argument entirely (throughout), or, perhaps the authors mean to say that they get these results without a (high-cost, slow, etc.) d-DNP polarizer? If so, say so.
- In the introduction, the authors make reference to past work with a related parahydrogen-based hyperpolarization technique, SABRE, in the context of efforts to continuously generate hyperpolarized molecules. Because SABRE does not chemically consume its substrate (by hydrogenation), any SABRE experiment that uses a bubbler can “continuously” generate hyperpolarization—albeit not on “new” substrates unless (e.g.) an additional (substrate, or substrate/catalyst) flow line is added to the reactor. In that sense, SABRE is relatively “easy” to make continuous or quasi-continuous. What the present authors have achieved here is more challenging, because there is the additional dimension of having to manage the chemical aspect of the PHIP reaction (conversion of reagent to HP product) and the subsequent conversion to ^{13}C hyperpolarization. So it reads strange to me that the authors are implying that continuous production hasn’t been performed well in SABRE yet (on the contrary, see, e.g., TomHon et al., CPC doi.org/10.1002/cphc.202100667, 2021) and that such a claim somehow is necessary to bolster the impact of the present work—whereas actually, the present work stands on its own (irrespective of what has been achieved with SABRE).
- Page 3: Text reads: “...The mean ^{13}C polarization was quantified with respect to a thermally polarized reference (5 mL acetone- ^{13}C , natural abundance ^{13}C fraction)...” What does that mean? Was the acetone ^{13}C labeled (isotopically enriched) or are the authors trying to say that they used

the ^{13}C NMR signal from the carbonyl resonance of neat acetone (with natural ^{13}C abundance)? Please clarify.

- In the paper (e.g. Fig. 1 caption) the performance is quoted as achieving: “PHP = (19 ± 1) % and PHP = (1.7 ± 0.2) %” respectively (standard deviation: dashed line).” However, the scatter in the spectra of Fig. 1, and the shifts of the dashed line positions in the insets, appear significantly larger than those claimed error bars. Those error bars likely need to be revised to be consistent with the shown data (with implications for the claimed quantitative reproducibility of the device).
- The paper refers to the interplay of different operation parameters affecting the hydrogenation yield (e.g. bottom of page 5) and achieved agent polarization. In practice, one would likely want (perhaps need) near-100% hydrogenation yield (at least for envisioned clinical applications—one would desire homogeneous, reproducibly specific chemical formulations for in vivo administration). What are the hydrogenation yields here, and how much do they vary from run to run (for a given setting of the device)? If the yields are not near 100%, then the authors should make clear to the uninitiated reader that the reported polarization values are for the successfully hydrogenated species only (as of course, any unreacted substrate will remain un-hyperpolarized).
- Page 6, first line of Discussion: “Producing four boli of hyperpolarized contrast agents per minute is at least two orders of magnitude faster than reported before.” However, it is unclear what comparison has been drawn here, so it’s hard to evaluate the validity of the claim. First, references should be provided. Second, greater specificity should be provided. For example, I’m *inferring* that the authors may be making a very limited, very specific claim (without saying so): i.e., that if one compares the rate of making exactly 4 batches here with the preparation of making exactly 4 (simultaneous) batches via d-DNP/SpinLab, then the present approach is two orders of magnitude faster (if so, I agree). However, certainly there have been many previous demonstrations of (at least individual) batches of the same or similar HP species via PHIP and SABRE with similar speed. Even if one had to manually re-load such batch preparation 4 times, surely the present approach would not be “two orders of magnitude faster”! Please make the present claim more specific/clear.

Very minor:

- Page 2, line 52, missing “a” before “dedicated polarizer device”
- Fig. 2, also page 5: what does SOT mean? Spin order transfer maybe?

Quasi-continuous production of highly hyperpolarized carbon-13 contrast agents every 15 s within an MRI system

Andreas B. Schmidt^{1,2,3,*}, Mirko Zimmermann¹, Stephan Berner^{1,2}, Henri de Maissin^{1,2}, Christoph A. Müller^{1,2}, Vladislav Ivantsev¹, Jürgen Hennig¹, Dominik v. Elverfeldt¹, Jan-Bernd Hövener^{3,*}

1 Department of Radiology, Medical Physics, Medical Center, University of Freiburg, Faculty of Medicine, University of Freiburg, Killianstr. 5a, Freiburg 79106, Germany.

2 German Cancer Consortium (DKTK), partner site Freiburg and German Cancer Research Center (DKFZ), Im Neuenheimer Feld 280, Heidelberg 69120, Germany.

3 Section Biomedical Imaging, Molecular Imaging North Competence Center (MOIN CC), Department of Radiology and Neuroradiology, University Medical Center Kiel, Kiel University, Am Botanischen Garten 14, 24118, Kiel, Germany.

E-Mail: andreas.schmidt@uniklinik-freiburg.de ; jan.hoevener@rad.uni-kiel.de

Pointwise response to the referees

We would like to acknowledge the reviewers efforts, very detailed analysis and feedback regarding our manuscript. While we are glad about the very positive evaluation of our work, we think that the comments raised helped us to avoid some confusion and to transport our results comprehensible.

We included all raised issues in the main text. In the following, we respond to the reviewers comments and report the included or edited text passages.

Kind regards,

Andreas Schmidt and Jan-Bernd Hövener

Reviewer R1:

The contribution by Schmidt et al. is an exciting step forward for the field of MRI using PHIP-polarized metabolites. The authors report on a new method to generate hyperpolarized biomolecules with a turnover time of 15s, which is close to continuous production. The achieved polarization levels are in line with the current state-of-the-art for the two molecules HEP and SUC, but the turnover rate is much higher. This opens the door to quasi-continuous or continuous hyperpolarized imaging.

Overall I think this work is well-suited for publication in Communications Chemistry, subject to a few minor corrections (see below).

Some specific comments to the authors:

R1.1:

In the introduction I think some more care should be given to introducing the current state-of-the-art in generating PHIP-polarized molecules in a quasi-continuous or continuous manner. The reader is not made aware of the current limitations in PHIP technology, and what advance has been made in this work. No mention is given to work in which many NMR tubes can be prepared containing frozen PHIP precursor solutions and used in quick succession (e.g., <https://doi.org/10.1016/j.jmr.2018.01.019>). Also, there is no explicit discussion of bubble-free PHIP setups which achieve continuous production of PHIP-polarized molecules (albeit at lower volume/concentration, which is worth discussing). See for example: <https://doi.org/10.1002/anie.201002725>, <https://doi.org/10.1021/jacs.9b03507>, <https://doi.org/10.1002/cphc.202100119> (which is already cited but lumped together with very different work).

A1.1:

Thanks for pointing this out. We agree and included a more detailed passage on the state of the art to our introduction:

“... ”

Parahydrogen (pH₂) induced polarization (PHIP) methods are drastically less expensive and much faster than other hyperpolarization (HP) schemes.^{15–18} There are two variants of PHIP: one where pH₂ is catalytically added to a precursor (referred to as hydrogenative PHIP, PASADENA^{15,19} or ALTADENA²⁰), and another where the substrate undergoes a reversible exchange with pH₂ at a catalyst and thus remains chemically unchanged (non-hydrogenative PHIP, SABRE).^{17,21,22} For the former, duty cycles of a few minutes were achieved^{23–25} – where the actual polarization transfer took less than 10 s at elevated temperatures and pH₂ pressures – e.g., by preparing several (frozen) samples in NMR tubes,^{26–31} or amounts of precursor-catalyst solution to be used in PHIP polarizers for production of agents polarized to >10 %.^{23,24,32–38} Such batches of PHIP-polarized carbon-13 CAs have recently been used for metabolic imaging with pyruvate⁸ and fumarate.³⁹

A very interesting development is continuous HP,⁴⁰ e.g., using excess of precursor and continuous pH₂ supply⁴¹ or continuous flow setups, e.g., to be used in lab-on-a-chip NMR devices.^{42–45} Only recently, continuous-flow PHIP of an acetate ester was demonstrated by bubble-free pH₂ dissolution and heterogeneous catalysis, which makes production of catalyst-free solutions straightforward.⁴⁶ SABRE is particularly well suited for continuous HP, because it does not consume the precursor.^{40,47–51} Progress is being made fast and continuous polarization of ¹H and X-nuclei (nuclei other than proton, e.g., ¹⁵N, ¹³C) in metabolites to ≈6 % and ≈2 %, respectively, was recently demonstrated in a bubble-free membrane reactor.⁵²

...”

R1.2:

Line 42: Duty cycle is the wrong parameter in this context. Turnover time (or similar) would be better.

A1.2:

With duty cycle, we refer to the total time needed to repeat the experiment, while turnover time often refers to the time needed to complete the hydrogenation. In line 42 we referred to work where the actual hyperpolarization (hydrogenation turnover + SOT) had been completed in <10s, and duty cycles have been reported in the range of few minutes. However, we agree that referring to the turnover / hyperpolarization time and the duty cycle in one sentence was confusing and hence, we tried to clarify:

“... ”

For the former, duty cycles of a few minutes were achieved²³⁻²⁵ – where the actual polarization, consisting of hydrogenation and spin order transfer (SOT), took less than 10 s at elevated temperatures and p_{H₂} pressures

...”

R1.3:

3. Line 64: “The deviation of the signals was likely dominated by the varying amount of filled reaction solution.” I do not understand this: why should this have such a large impact on polarization, and why is there such a significant standard deviation on the amount of reaction solution being used? I think this point certainly needs clarifying, since the standard deviation of polarization looks close to 20% in both cases.

A1.3:

Right, this for sure requires clarification, thanks. We added a paragraph to the main text describing and discussing how we quantify the hyperpolarization. A difference to many other PHIP setups is that we quantify P_{13C} in the MRI system from two non-localized (global) NMR spectra (one from the polarized sample and one from a thermal reference) using a large volume coil. Thus, the measured signals are acquired from the complete sample - in contrast, for instance, in NMR spectrometers the small receive coil records the signal only from a sensitive volume that covers a fraction of the sample. As a consequence, a variation in the sample volume does change the measured hyperpolarized signal. The sample volume varies for two reasons; first, the injection of pressurized parahydrogen flushes out a variable amount of solution from the reactor (and from the shimmed B₀ and sensitive B₁ field); second, we additionally observed some variation in volume caused by the fast injection of unreacted precursor solution into the reactor. Of course, the actual sample volume could be measured, *e.g.*, with 1H MRI, but only at the cost of duty cycle. Hence, for the quantification we assume 700µL of solution that was left after hyperpolarization in the reactor on average.

We added the following text to our manuscript:

“... ”

These numbers require some commenting. First, the polarizations are likely underestimated as we assumed complete hydrogenation, whereas only $\approx 90\%$ were actually achieved (see below); complete hydrogenation would increase the polarization by a factor of ~ 1.11 . Secondly, using p_{H₂} enriched to 100% (instead of 85% used here) would increase the polarization by another factor of ≈ 1.25 . Thirdly, we note that the standard deviation of rapid-PHIP ($\pm 20\%$) is larger than for single-batch SAMBADENA ($\pm 10\%$). We attribute this variation in P_{13C} to varying volumes of CA in the reactor, whereas we assume a constant volume (700 µl) for

quantification. We observed some variation in volume caused by the fast injection of unreacted precursor solution into the reactor, and to some spill over during the hydrogenation reaction.

...”

R1.4:

4. Line 78: I think there are suitable pulse sequences for this such as S2hM (<https://doi.org/10.1016/j.jmr.2017.03.002>) and even generalized-S2hM (<https://doi.org/10.1016/j.jmr.2020.106850>). I don't insist on repeating experiments using these pulse sequences, but they could be mentioned as viable alternatives.

A1.4:

Agreed. We added the following text:

“ ...

Note that polarization for HEP could be doubled (*i.e.*, to $\approx 40\%$ for ^{13}C) if a SOT method would be used that provides up to 100 % polarization instead of 50 % - a matter currently being investigated.^{60,68,69} Also for SUC, other SOT sequences (*i.e.*, for strongly-coupled spin systems) seem to be well-suited like the (generalized) S2hM sequence and should be tested in future experiments.^{53,54,59,70-75}

...”

R1.5:

5. Line 127: I take issue with the statement “Producing four boli of hyperpolarized contrast agents per minute is at least two orders of magnitude faster than reported before.”. Many PHIP laboratories are able to repeat experiments faster than once every 25 minutes. This is even stated on line 42 in the Introduction. I think one order of magnitude is more accurate.

A1.5:

Thank you, we report this more clearly in the manuscript now. As you have suggested, we introduce a more detailed PHIP state-of-the-art (see A1.1). Then, the first sentence of the discussion was changed the following:

“ ...

Producing four boli of hyperpolarized contrast agents per minute (or 10 in 2.5 min) as demonstrated here is about two orders of magnitude faster than reported before with dDNP. *E.g.*, one sample every hour has been achieved,¹³ and two to four simultaneously-polarized batches have been used subsequently every 12 min after ≈ 3 h of microwave irradiation followed by a 30-h recovery period.¹² Compared to duty cycles of minutes reported for hydrogenative PHIP to produce HP batches of $P_{^{13}\text{C}} > 10\%$, a 15-s duty cycle is still several times faster, *e.g.*, by a factor of ≈ 12 ³⁶ or ≈ 20 .²³

...”

Reviewer R2:

This is a review for a manuscript recently submitted by Schmidt et al. entitled “Quasi-continuous production of highly hyperpolarized carbon-13 contrast agents every 15 s within an MRI system”, submitted for consideration for publication in Communications Chemistry.

Ultimately, I believe the paper will be publishable once some relatively minor issues and questions have been addressed (mostly just concerning how the authors frame some of their advances), given that most of the science itself seems to be in good order (and the paper for the most part is clearly written).

In their Communication, the authors describe recent work to develop a rapid and reproducible approach for creating select hyperpolarized ^{13}C -labeled agents for biomedical imaging applications via parahydrogen-induced polarization (PHIP). The approach exploits methodologies previously developed by the group, largely under the acronym SAMBADENA--a PHIP process that occurs within an MRI scanner (facilitating the rapid transfer of hyperpolarized agents to the target/subject with minimal relaxation losses). In addition to rapid agent production, good reproducibly and relatively high ^{13}C polarizations are obtained by optimizing this process (e.g. $P(^{13}\text{C}) \sim 2\%$ for succinate or $\sim 19\%$ for hydroxyethyl-propionate, with batches every 15 s) with rational experimental design centered around a dedicated PHIP reactor, which should ultimately help enable PHIP-enhanced metabolic MRI with continuous (or at least quasi-continuous) HP agent administration.

Specific comments / questions / corrections (in rough order of the manuscript):

R2.1:

- In many places, the authors say that they achieve their results “without a polarizer”. This strikes me as a strange claim: (1) the authors make it sound like this is somehow an advantage, but it’s not clear to me why this would be an advantage as one must have *some* way to make HP agents; and (2) in fact, I would say that the authors do indeed have a “polarizer”. The authors describe a dedicated heated/controlled PHIP reactor with additional controls on the front and back end (including integration with the MRI scanner), such that into it goes pH_2 and substrate, and out comes HP agent (quite nicely). Perhaps it does not have a fully enclosed cabinet and wheels, but besides that, it sounds like a PHIP polarizer to me! I suggest the authors either delete this line of argument entirely (throughout), or, perhaps the authors mean to say that they get these results without a (high-cost, slow, etc.) d-DNP polarizer? If so, say so.

A2.1:

Thank you! We meant that we use no dedicated stand-alone polarizer device but a setup integrated to the MRI instead. We changed this in all instances, for instance in the abstract:

“ ...

is demonstrated within an MRI system without a stand-alone polarizer, but with a setup integrated in an MRI instead.

...”

R2.2:

- In the introduction, the authors make reference to past work with a related parahydrogen-based hyperpolarization technique, SABRE, in the context of efforts to continuously generate hyperpolarized molecules. Because SABRE does not chemically consume its substrate (by hydrogenation), any SABRE experiment that uses a bubbler can “continuously” generate hyperpolarization—albeit not on “new” substrates unless (e.g.) an additional (substrate, or substrate/catalyst) flow line is added to the reactor. In that sense, SABRE is relatively “easy” to make continuous or quasi-continuous. What the present authors have achieved here is more challenging, because there is the additional dimension of having to manage the chemical aspect of the PHIP reaction (conversion of reagent to HP product) and the subsequent conversion to ^{13}C hyperpolarization. So it reads strange to me that the authors are implying that continuous production hasn’t been performed well in SABRE yet (on the contrary, see, e.g., TomHon et al., CPC doi.org/10.1002/cphc.202100667, 2021) and that such a claim somehow is necessary to bolster the impact of the present work—whereas actually, the present work stands on its own (irrespective of what has been achieved with SABRE).

A2.2:

Thank you for this important remark! We for sure did not intend to disregard achievements made with hydrogenative PHIP and SABRE at all and indeed, are aware of the rapid and impressive progress being made for both variants. We rewrote the introduction of the PHIP state of the art and gave more emphasis on previous developments and achievements (see answer to reviewer 1, A1.1).

R2.3:

- Page 3: Text reads: “...The mean ^{13}C polarization was quantified with respect to a thermally polarized reference (5 mL acetone- $2\text{-}^{13}\text{C}$, natural abundance ^{13}C fraction)...” What does that mean? Was the acetone ^{13}C labeled (isotopically enriched) or are the authors trying to say that they used the ^{13}C NMR signal from the carbonyl resonance of neat acetone (with natural ^{13}C abundance)? Please clarify.

A2.3:

We changed the sentence to:

“... ”

The mean ^{13}C polarization was quantified with respect to a thermally-polarized reference (carbonyl resonance of 5 mL neat acetone, natural abundance ^{13}C of $\approx 1.1\%$)

“... ”

R2.4:

- In the paper (e.g. Fig. 1 caption) the performance is quoted as achieving: “PHP = $(19 \pm 1)\%$ and PHP = $(1.7 \pm 0.2)\%$ ” respectively (standard deviation: dashed line).” However, the scatter in the spectra of Fig. 1, and the shifts of the dashed line positions in the insets, appear significantly larger than those claimed error bars. Those error bars likely need to be revised to be consistent with the shown data (with implications for the claimed quantitative reproducibility of the device).

A2.4:

Thank you! We report mean values with the standard error of the mean (which is \sqrt{N} -fold smaller than the standard deviation) but decided that in the figure, the standard deviation suits better. We clarified this issue:

Main text:

“ ...

to $P_{HP}(HEP) = (19 \pm 1) \%$ and $P_{HP}(SUC) = (1.7 \pm 0.2) \%$ (mean \pm standard error; the standard deviation in absolute numbers was $\pm 4 \%$ and $\pm 0.5 \%$, respectively; Fig. 1).

...”

Caption Figure 1:

“ ...

The mean ^{13}C polarization (right side, red line) with standard error, s_E , was $P_{HP} = (19 \pm 1) \%$ and $P_{HP} = (1.7 \pm 0.2) \%$, respectively (standard deviation of all N experiments, $\sigma = \sqrt{N} \cdot s_E$, plotted as red dashed line).

...”

R2.5:

- The paper refers to the interplay of different operation parameters affecting the hydrogenation yield (e.g. bottom of page 5) and achieved agent polarization. In practice, one would likely want (perhaps need) near-100% hydrogenation yield (at least for envisioned clinical applications—one would desire homogeneous, reproducibly specific chemical formulations for in vivo administration). What are the hydrogenation yields here, and how much do they vary from run to run (for a given setting of the device)? If the yields are not near 100%, then the authors should make clear to the uninitiated reader that the reported polarization values are for the successfully hydrogenated species only (as of course, any unreacted substrate will remain un-hyperpolarized).

A2.5:

Thank you very much for this comment – an issue that we had left undiscussed here. The precursor concentration can be reduced with a purification or a more completed (longer or faster) hydrogenation. We rewrote the corresponding main text and added a long paragraph to address this matter more precisely (new text highlighted):

“ ...

Not surprisingly, we found that the reaction temperature was beneficial for the HP yield, just as well as the $p\text{H}_2$ pressure (Fig. 3a,b^{24,35} – note that the hydrogenation yield was not considered in the quantification, but assumed to be 100 %). The catalyst concentration did not have a pronounced effect on the polarization in the range of 1-4 mM for 80 mM of HEA before (Fig. 3c).⁵⁵ As expected, the ^{13}C polarization decreased and the payload (precursor concentration multiplied by polarization) reached a maximum when the concentration of the precursor hydroxyethyl [1- ^{13}C]acrylate- d_3 (HEA), c_{HEA} , was increased from 5 mM to 80 mM under otherwise same conditions (Fig. 3d). A mono-exponential saturation function fitted the payload (d) as function of c_{HEA} well, suggesting that one other reactant ($p\text{H}_2$, or catalyst) was used. Together with (b) and (c), this observation suggests that $p\text{H}_2$ was likely the rate limiting reactant here, and that the saturation of the payload in Fig. 3d was attributable to the consumption of the available $p\text{H}_2$. Prolonging the hydrogenation period increased the hydrogenation yield, but caused relaxation at the same time.³⁵ Thus, there was an optimum hydrogenation time, t_h , of 5 s for which the polarization, under the chosen parameters, was maximal (Fig. 3e).

In some cases, however, reducing the amount of unreacted precursor may be desired, in particular if it is toxic. While this could be addressed with a purification of the solution, e.g., with precipitation,^{38,76} a longer or faster hydrogenation may solve this issue, too. To investigate the matter further, a two-handed kinetics model has been fitted to the polarization as function of t_h (Fig. 3e):^{31,35,54,77}

$$P_{13\text{C}}(t_h) = P_{\text{max}} / |T_{\text{hydr}}/T_{\text{relax}} - 1| \cdot [\exp(-(t_h - t_0)/T_{\text{hydr}}) - \exp(-(t_h - t_0)/T_{\text{relax}})]$$

Where T_{hydr} and T_{relax} are time constants of the hydrogenation and relaxation of the ^1H spin order, respectively, P_{max} is the maximum polarization value (reached if no relaxation were present), and t_0 is a time offset (delivery and dissolution of $p\text{H}_2$, build-up of pressure, and activation of catalyst). The fitted parameters were: $T_{\text{hydr}} = (1.6 \pm 0.9) \text{ s}$, $T_{\text{relax}} = (16 \pm 6) \text{ s}$, $P_{\text{max}} = (27 \pm 5) \%$, and $t_0 = (1.0 \pm 0.5) \text{ s}$. With $f(t_h; t_0, T_{\text{hydr}}) =$

$\exp(-(t_h - t_0)/T_{\text{hydr}})$, the reacted fraction, f , with propagated error, s_f , after hydrogenation time t can be calculated as

$$f \pm s_f = f(t_h; t_0, T_{\text{hydr}}) \pm \left(\left(\frac{\partial}{\partial t_0} \cdot f(t; t_0, T_{\text{hydr}}) \right)^2 \cdot s_{t_0}^2 + \left(\frac{\partial}{\partial T_{\text{hydr}}} \cdot f(t; t_0, T_{\text{hydr}}) \right)^2 \cdot s_{T_{\text{hydr}}}^2 \right)^{1/2}$$

Hence, the model suggested that after 5 s, (92 ± 12) %, and after 10 s, (99 ± 1) % of the precursor were reacted (under the given conditions of $p_{\text{pH}_2} = 15$ bar, $c_{\text{cat}} = 2$ mM, $c_{\text{HEA}} = 5$ mM).

Combined, these findings suggest that a faster reaction and an increased p_{H_2} availability would improve the hydrogenation and allow to effectively polarize higher concentrations (e.g., 80 mM, Fig. 3d). Increasing the pressure is one solution that we are currently pursuing. Another one is to use solvents with higher p_{H_2} solubility than water (≈ 0.8 mM / bar at 25°C)^{78,79}, for instance chloroform (≈ 2.7 mM / bar at 25°C)⁸⁰ or acetone (≈ 3.9 mM / bar at 25°C)⁸¹. Note that these solvents are typically used for PHIP by side arm hydrogenation (PHIP-SAH)⁸² for production of metabolic PHIP agents, which makes this approach most promising, too. Interestingly, the solubility of H_2 in chloroform and acetone increases with temperature (e.g., to 3.4 and 4.6 mM / bar at 50°C , respectively), whereas it decreases for water (0.7 mM / bar at 50°C).^{78,80,81,83}

Figure 3| Optimization of rapid PHIP. ...

...”

R2.6:

- Page 6, first line of Discussion: “Producing four boli of hyperpolarized contrast agents per minute is at least two orders of magnitude faster than reported before.” However, it is unclear what comparison has been drawn here, so it’s hard to evaluate the validity of the claim. First, references should be provided. Second, greater specificity should be provided. For example, I’m *inferring* that the authors may be making a very limited, very specific claim (without saying so): i.e., that if one compares the rate of making exactly 4 batches here with the preparation of making exactly 4 (simultaneous) batches via d-DNP/SpinLab, then the present approach is two orders of magnitude faster (if so, I agree). However, certainly there have been many previous demonstrations of (at least individual) batches of the same or similar HP species via PHIP and SABRE with similar speed. Even if one had to manually re-load such batch preparation 4 times, surely the present approach would not be “two orders of magnitude faster”! Please make the present claim more specific/clear.

A2.6:

We added more specific numbers and citations and edited the sentences the following:

“... ”

Producing four boli of hyperpolarized contrast agents per minute (or 10 in 2.5 min) as demonstrated here is about two orders of magnitude faster than reported before with dDNP. *E.g.*, one sample every hour has been achieved,¹³ and two to four simultaneously-polarized batches have been used subsequently every 12 min after ≈ 3 h of microwave irradiation followed by a 30-h recovery period.¹² Compared to duty cycles of minutes reported for hydrogenative PHIP to produce HP batches of $P_{13C} > 10\%$, a 15-s duty cycle is still several times faster, *e.g.*, by a factor of ≈ 12 ³⁶ or ≈ 20 .²³

...”

Very minor:

R2.7:

- Page 2, line 52, missing “a” before “dedicated polarizer device”
- Fig. 2, also page 5: what does SOT mean? Spin order transfer maybe?

A2.7:

Thank you! We added the suggestions.

REVIEWERS' COMMENTS:

Reviewer #1 (Remarks to the Author):

The authors have responded well to all comments and I consider this manuscript to be acceptable in its current form for publication in CommsChem.

Reviewer #2 (Remarks to the Author):

I am satisfied with the changes to the manuscript that the authors have made as part of their responses to the reviewer comments. I recommend publication.

Quasi-continuous production of highly hyperpolarized carbon-13 contrast agents every 15 seconds within an MRI system

Andreas B. Schmidt^{1,2,3,*}, Mirko Zimmermann¹, Stephan Berner^{1,2}, Henri de Maissin^{1,2}, Christoph A. Müller^{1,2}, Vladislav Ivantsev¹, Jürgen Hennig¹, Dominik v. Elverfeldt¹, Jan-Bernd Hövener^{3,*}

1 Department of Radiology, Medical Physics, Medical Center, University of Freiburg, Faculty of Medicine, University of Freiburg, Killianstr. 5a, Freiburg 79106, Germany.

2 German Cancer Consortium (DKTK), partner site Freiburg and German Cancer Research Center (DKFZ), Im Neuenheimer Feld 280, Heidelberg 69120, Germany.

3 Section Biomedical Imaging, Molecular Imaging North Competence Center (MOIN CC), Department of Radiology and Neuroradiology, University Medical Center Kiel, Kiel University, Am Botanischen Garten 14, 24118, Kiel, Germany.

E-Mail: andreas.schmidt@uniklinik-freiburg.de ; jan.hoevener@rad.uni-kiel.de

Pointwise response to the referees

REVIEWERS' COMMENTS:

Reviewer #1 (Remarks to the Author):

The authors have responded well to all comments and I consider this manuscript to be acceptable in its current form for publication in CommsChem.

Reviewer #2 (Remarks to the Author):

I am satisfied with the changes to the manuscript that the authors have made as part of their responses to the reviewer comments. I recommend publication.

AUTHOR RESPONSE:

We would like to acknowledge the reviewers time and efforts and are glad that we satisfied and answered the issues raised in the previous revisions.

Many thanks and kind regards,

Andreas Schmidt and Jan-Bernd Hövener